# Children’s Participation in Free School Meals: A Qualitative Study among Pupils, Parents, and Teachers

**DOI:** 10.3390/nu14061282

**Published:** 2022-03-18

**Authors:** Sandra Mauer, Liv Elin Torheim, Laura Terragni

**Affiliations:** Department of Nursing and Health Promotion, Faculty of Health Sciences, Oslo Metropolitan University (OsloMet), 0130 Oslo, Norway; sandra.mauer.no@gmail.com (S.M.); livtor@oslomet.no (L.E.T.)

**Keywords:** school meals, nutrition policies, food preferences, healthy meals, children, Norway, qualitative methods

## Abstract

School meals can have a key function in promoting children’s health. However, simply providing a free school meal is not a guarantee that pupils will eat the food. The purpose of this study was to explore factors influencing pupils’ participation in free school meal schemes in Oslo. The study has a qualitative research design, inspired by grounded theory. Data were collected through interviews with pupils, teachers, and parents, and participant observations in two schools participating in a pilot project funded by Oslo Municipality. Line-by-line coding, memo writing, and a constant comparative technique were used to analyze the data. One primary school and one lower-secondary school in different districts in Oslo that were implementing two different free school meal models took part in the study. In total, 39 pupils (5th–10th grade), 15 parents, and 12 school employees were included. Four main factors related to pupils’ participation in free school meals emerged from the analysis: the popularity of the food served, the attraction to the nearby shopping center, social aspects, and predictability. To promote pupils’ participation in free school meal schemes, schools need to solve the challenges of balancing between healthy food and popular but often unhealthy food. To implement school meals further, children and parents’ involvement, regularity of the meals provision, a good flow of information, and the creation of a friendly eating environment are recommended.

## 1. Introduction

Children and adolescents spend many hours every day at school. Therefore, schools are recognized as an arena with the potential to promote healthy habits in children and adolescents and to reduce social differences in diets [1,2,3]. 

While some of the Nordic countries such as Sweden and Finland have implemented a national free school lunch program for a long time, in Norway, school meal provision is not statutory, and it is common for the pupils to eat a bread-based packed lunch brought from home [4,5]. There is, however, a state-funded support scheme for fruit and vegetables and school milk in primary and secondary schools [5,6,7]. 

Currently, about 16% of 13-year-old Norwegian adolescents are overweight or obese [8]. Norwegian school children, in general, consume too much saturated fats and foods and drinks with added sugar, and they eat too little fruits, vegetables, and fish in relation to recommendations [9,10,11,12]. Concerns about pupils’ meal patterns have also been raised as many Norwegian adolescents skip breakfast [11,12,13,14] or do not eat the packed lunches they bring from home but instead buy unhealthy snacks and sugary drinks [12,13,15,16]. Social differences in adolescents’ food consumption have been documented: children from families with high socioeconomic status (SES) have a healthier diet than children with low SES [10,13,17,18]. 

Food choices and physical activity can be changed towards healthier behavior, especially when learned from childhood [19]. This can lead to a healthier lifestyle in adults and reduce the risk of obesity and related illnesses [20]. Therefore, children are an important target group for early public health promotion interventions [21,22].

A recent systematic review confirmed that universal free school lunch was positively associated with school meal participation, diet quality, academic performance, and food security [23]. Both randomized controlled trials (RCTs) and longitudinal observation studies in Scandinavia have shown the potential of free school meals and free fruits and vegetables at school to improve pupils’ diets [23,24,25,26,27] and reduce socioeconomic inequalities [28,29]. In recent years, interest in school meals both globally and in Norway has increased. In Norway, there have been several initiatives and pilot projects to test different school meal models and how school meals could be implemented [30,31]. The few intervention studies that have been conducted have generally shown an association with a healthier diet [28,32,33,34,35,36].

However, simply providing a free meal is not a guarantee that pupils will participate in the program and eat the food [37], and several challenges related to the practical implementation of school meals have been identified. These challenges include meeting children’s food preferences [38,39], social challenges such as social acceptability and stigmatization, and pupils’ eating autonomy [40,41,42]. The food environment near schools, such as fast-food outlets and grocery stores, may also pose challenges to children’s participation in school meals [43]. 

In 2019, the Agency for Health in Oslo started a two-year pilot project (2019–2021) of supporting free school meals at the primary and lower-secondary levels to follow up Oslo City Council’s decision to introduce free healthy school meals to children in Oslo to improve public health [44]. This is particularly relevant in a city such as Oslo, where health inequality is a major public challenge [45,46]. 

The aim of this study was to gain a better understanding of what influences pupils’ participation in free school meal schemes. Specifically, we wanted to explore pupils’, school staff, and parents’ experiences with and views on free school meals introduced at two of three pilot schools in Oslo.

## 2. Materials and Methods

### 2.1. Study Design

The study was informed by grounded theory (GT) [47], which allows researchers permission to start data collection and develop a new theory out of the data [48]. This approach was regarded as suitable as the project started without predetermined assumptions and aimed at exploring potential influencing factors and the social processes underpinning the school meals phenomenon [49]. A total of 19 qualitative group interviews and one in-depth interview with pupils, teachers, and parents were conducted between October 2020 and January 2021 (Table 1). In addition, participant observations were undertaken during breakfast (primary school), and food production and lunch (lower-secondary school). The Norwegian Center for Research Data (NSD) gave ethical approval for this study (reference number: 598231). 

### 2.2. Recruitment and Sample

The three schools involved in Oslo’s pilot project were invited to participate in the present study, and two agreed, a primary and a secondary school. The primary school served a breakfast buffet four days a week. The lower-secondary school served lunch twice a week with the menu varying weekly and consisting of both hot and cold dishes. A food and health teacher (canteen manager) and pupils from the elective subject ‘Production of goods and services’ prepared and served the school lunches. Both schools were large multicultural community schools placed in neighborhoods with a high proportion of immigrants and lower SES. The primary school (1st –7th grade) had 524 pupils, whereas the lower-secondary school (1st–10th grade) had 593 pupils. The pupils were recruited through their teachers that informed them about the project. In total 47 pupils were interested and 39 participated in the study. The school staff was chosen among those having practical experience with the school meals project. In total 13 informants among the staff were contacted and 12 participated in the study. Parents were invited to participate by the schools’ administrations, from whom they received an information letter by e-mail. In total 660 parents were invited and 15 participated. Signed informed consent forms were obtained from participating pupils’ parents before data collection.

Following GT, we adopted a theoretical sampling aiming to include individuals with different experiences and perspectives on the phenomenon being studied. Three groups at each school were invited to participate in the study: pupils (primary school: 5th–7th grade; lower-secondary school: 8th–10th grade), school staff (teachers, principals, canteen managers/cooks) and parents. In the lower-secondary school, we interviewed two groups of pupils: one that was involved in the meal preparation (as part of an elective subject) and another consisting of representatives from the pupils’ council. 

### 2.3. Data Collection

To better understand different aspects and obtain a collective view of the free school meals through organized discussions, we initially planned to conduct focus group interviews. However, because of COVID-19, only the first four interviews were conducted in person, while the remaining were conducted digitally over Zoom (web-based video conferencing). To handle the participants more easily, a maximum of 2–3 participants was invited per group interview via Zoom. The prerequisite for focus group sizes is approximately 6–8 participants [50], but this was mainly not met as the number of participants in 16 out of 19 group interviews was less than four. For this reason, we use the term ‘group interviews’ [51].

Nineteen group interviews and one in-depth interview with 66 participants were undertaken. The one in-depth interview was conducted as the parent could not attend the focus group but wanted to participate in the study. These included eight group interviews with pupils (*n* = 39), seven group interviews with parents (*n* = 15), and five interviews with school staff (*n* = 12) (Table 1). 

Three days of observation during school breakfast at the primary school and five days of observation of food production and lunch at the lower-secondary school were conducted before the interviews. Observation included the place where the meals were served, what was served and interactions occurring during the meals or more generally during lunch breaks in the schoolyard. Reflections from the observations were used to develop the interview guides. The semi-structured interview guides for the group interviews were developed in line with a GT approach, pilot tested, and revised before new interviews were conducted as the need for adding new topics or shifting some of the focus of the investigation emerged. For instance, one additional question was added after some primary school pupils spoke of having to get up earlier for the school breakfast. The group interviews contained questions about the school meals, what participants experienced as positive and negative, the eating environment and pupils’ and parents’ involvement. The interviews lasted 30–60 min and were audio-recorded and transcribed verbatim shortly after they were conducted. 

### 2.4. Data Analysis

The analysis was based on the interview transcripts, the schools’ project descriptions and memos. NVivo 12 (qualitative data analysis software) was utilized in the data analysis and theoretical development. 

Alongside the data collection, the first author started with an initial analysis of the interviews to adjust the interview guides and proceed with further recruitment. Following GT, the analysis was conducted in two phases of initial coding and focused coding [47]. In the first phase, line-by-line coding was used in the first interviews, and gradually segments or smaller sections were coded for all the remaining interviews. The codes in the data were continuously compared to data from new interviews, looking for similarities and differences. When new codes emerged from the data, previous interviews were re-read, searching for the same patterns. Then, focused coding was used to categorize the data. For instance, initial codes as ‘does not want to eat alone’ and ‘importance of eating with someone’, were then grouped under the category ‘social eating’. The first author developed a preliminary codebook, which was then discussed with the other authors and staff from the Agency for Health in Oslo. While doing the focused coding, memos were written to identify interesting thoughts and aspects that could lead to emerging themes in the data. By reading and sorting codes and creating categories, all data were reviewed several times, and mind-mapping was used to foster theoretical thinking around the data and draw connections between categories. The final codebook was reached by sorting the codes and categories and by working with mind mapping. 

## 3. Results

The two schools opted for two different meal programs. The primary school served breakfast, while the lower-secondary school provided lunch. Breakfast was served from 8 am to 8.25 am, in the school’s assembly hall. It was a quiet and cozy environment. The observations showed that the breakfast buffet included whole meal bread, but also polar bread (Scandinavian soft flatbread), various toppings such as jams in portion packs (strawberries and blueberries), mackerel in tomato sauce, caviar (smoked cod roe in tubes), chicken pâté in portion packs, cold cuts (pastrami chicken halal-sausage, salami), cheese slices, homemade chicken salad, mayonnaise and sliced fruit and vegetables (apple, orange, grapes, carrot), homemade “granola” and bowls with ‘Honey Grains’ and ‘Cheerios’ as breakfast cereals. There was also milk, orange juice, apple juice, water, and a bowl of vanilla yogurt.

Lunch was served in the auditorium. Both pupils eating the free lunch and those eating food brought from home could sit there and the place could become crowded and a bit noisy. The dishes served for lunch during the school meal project were, for example, tacos, au gratin cheese sandwiches, chili con carne, chicken salad, pasta, meat dishes, quesadillas, fish burgers, hamburgers, tomato soup, fish soup, salad bar, smoothie bowls, yoghurt with fruit and berries, and juice (sugar-free).

The findings indicate that there were several influencing factors for participating in the free school meals and that they tended to be interconnected. The most important factors were the popularity of the food served, competition with the nearby shopping center, social aspects, and predictability. 

### 3.1. Popularity of the Food

Popularity of the food was of utmost importance for attracting pupils to the school meals. According to pupils, popular foods were those that many were attracted to and that were not usually served at home daily. Toast, pancakes, and cereals such as ‘Cheerios’ were examples of ‘highlights’ for breakfast. Other popular breakfast dishes were cornflakes, yoghurt and polar bread. Tacos, au gratin cheese sandwiches, chili con carne, pasta, meat dishes, quesadillas, burgers, smoothie bowls, and yoghurt with fruits and berries were mentioned by the pupils and the canteen manager at the secondary school as popular lunch dishes. Fish, vegetarian food, and salads were instead often mentioned as unpopular. The following dialogue during a group interview with children involved in meal preparation exemplifies definitions of ‘popular and unpopular food’.


*Int. So, what kind of food is most popular?*



*The kind of food that is not healthy in a way. Pasta, for instance. But it depends on what kind of sauce you add. And then meat, chicken. The meals with meat are very popular and many come to eat.*
(Boy, 9th grade, GN1)


*Int. Ok, and what is unpopular?*



*Fish (many laugh) Fish can be good if you mix it with something else. But is never good alone.*
(Boy, 9th grade, GN1)

The interviews indicated that pupils’ and school staff’s ideas of ‘good food’ to serve are not necessarily the same. For instance, the new chef in charge of breakfast regarded the most popular options (e.g., toast and pancakes) as not that healthy and removed them from the breakfast buffet with a few exceptions. This seems to have had an influence on participation:


*I agree that [the classmates] came because of the toast (…) but now it is just ordinary food that you can just as well eat at home.*
(Boy, 7th grade, GN13)


*One day we got just pancakes, and we thought that we were going to get pancakes every day. So, we said to the others: there are pancakes, you have to come! But we were wrong. The next week many came to the breakfast, but there were no pancakes. So they just turned around and went away.*
(Girl, 6th grade, GN12)

The pupils who produced the lunch meals also found it was challenging to find a good balance between cooking healthy food and serving popular food: 


*We had a survey, where people should write what they would like to have. Then there came up a lot of pancakes, waffles, smoothie bowls and things like that. (…) We cannot get all our wishes fulfilled (…) because we are not allowed because of Oslo Municipality. They sponsor us with healthy food, or we have to buy healthy food. (…) Yes, and then it will be less popular.*
(Girl, 10th grade, GN2)

An indication that the food was not popular was when it was not eaten or was thrown away. For example, sometimes the chicken was picked out and eaten from the chicken salad, while the salad was thrown away: 


*I think a lot of food is thrown away because, for instance, when chicken salad is served, they [the classmates] are eating the chicken. But they do not always eat the salad. Thus, you take the salad because of the chicken.*
(Boy, 9th grade, GN1)


*If they do not like it [the lunch], then they throw it right away. If it tastes bad in any way, they go to the bathroom and throw it away.*
(Girl, 10th grade, GN2)

The canteen manager argued for the importance of finding a balance between healthy and unhealthy food, and the pupils gave suggestions for producing popular food such as burgers in a healthier way: 


*As I said to you, cabbage soup that nutritionally would be very good, it would be just you and me who ate it. So, I have to find—you have to find—dishes which appeal to that age group.*
(Teacher, GN4)


*Burgers are usually unhealthy, but today for example, we had wholegrain bread and fish instead of meat.*
(Boy, 9th grade, GN1)

### 3.2. Competition with Other Options: The Temptation of the Nearby Shopping Center 

Pupils from 8th to 10th grade were allowed to leave the school yard during the lunch break. The lower-secondary school had a shopping center nearby, and this resulted in a competition between the school canteen and food outlets at the center. The attraction of the shopping center seemed to increase with age:


*Yes, in the autumn of the 8th grade, all the pupils bring a packed lunch. (…) Eh, and then the visits to the center increases as the lunch boxes disappear. Eh, so it is true that the 10th grade has traditionally been a lot at the center. They still do it now, even though they are offered [free] food.*
(Teacher, GN8)

The reasons pupils and teachers mentioned for going to the center were that the pupils had not brought a packed lunch, that they wanted some snacks or something yummy or simply to follow their friends: 


*It really depends on the day, whether it’s a day where you feel like having bread or rather a small meatball, or something like that. Otherwise, it’s a day when you feel like having something sweet and just buy chocolate, you know.*
(Girl, 10th grade, GN2)

The teachers reported energy drinks as a main attraction in addition to what the oldest pupils stated buying:


*There are a lot of sweet buns and energy drinks. They [the 10th graders] buy it. That’s what they go to the center for buying. And it has something to do with age. They are allowed to buy energy drinks when they have become 10th graders, and then they can buy it. And especially the boys, not so many girls but many boys are buying energy drinks every day.*
(Teacher, GN8)

Some of the parents also mentioned that the shopping center has a powerful appeal. As one mother explained: 


*I think the older you get in lower-secondary school, the more you want to detach yourself from maybe being a pupil in primary school. (…) I don’t think this is strange—that they try to free themselves more and more from having to eat at school and so on when they become older and when they reach the 10th grade.*
(Mother, GN5)

### 3.3. Eating Food Together: The Sociality of Meals at School

Another emerging theme for participating in school meals was eating and being social with friends. At the primary school, many of the pupils and parents stated that the pupils enjoyed eating breakfast with their friends more than having breakfast at home alone as in many cases, parents left for work early, or were still sleeping: 


*I think the breakfast is really good. And I usually attend it. Sometimes I’m also there with lots of friends and stuff, so it’s really fun.*
(Boy, 7th grade, GN14)


*Because in the morning it is, like we as a family we are getting up at different times, so that [the children] often eat breakfast alone. So, therefore, [the children] thought it was all right to eat with someone.*
(Mother, GN16)

Additionally, in the lower-secondary school, both the pupils and school staff described the days with school lunch as more social than other days. 


*If one or two are sitting and eating food from the canteen, all the other friends will be there as well and sit together. So there are more people in the auditorium. And you hear it very well, and it is more social. And there are fewer who go to the center, I think.*
(Girl, 9th grade, GN1)


*[…] there is a different kind of unity. We notice that. They say that when there is food [for lunch], they stay there. And then they sit there and talk and game instead of going to the center.*
(Mother, n6, GN6)

The school meals seemed to be an opportunity for adults and pupils as well to meet at an arena other than the classroom, helping building relations both among the pupils and between pupils and adults:


*But it’s kind of, the young teachers who tend to be at the school breakfast. Yes, it’s a bit; you are allowed to joke a bit with them then.*
(Boy, 7th-grade, GN14)


*Eh, sometimes the principal or some of the teachers come, and then they come and eat with us and talk to us like whether we’re fine and so on.*
(Girl, 7thgrade, GN13)

The influence of friends and classmates in participating in the school meals emerged when observing that there was variation in participation according to class: in some classes, many pupils participated in the free school meal, while in others, most of the pupils did not: 


*Maybe [the classmates] think that if my friend isn’t coming, then I do not bother, and then also the others do not bother to come.*
(Boy, 7th-grade, GN14)


*I think this quickly sort of becomes a bit like part of the culture in a class, you know. I saw that when my eldest did not go [to the school breakfast], I don’t think there was such a large attendance from the whole class at the time. And that it was rather maybe a bunch of girls but none of the boys, and that I think is just a bit like a group mentality then. (…), it is difficult perhaps to turn it around a bit because then it is only defined as not cool somehow.*
(Mother, n11, GN15)


*We have a large group of boys [10th grade] who do not eat in the canteen, period.*
(Teacher, GN8)

### 3.4. Predictability and Continuity of the Provided Meals

In this pilot project, school meals were not served every day, as breakfast was served four days a week and lunch twice a week. In addition, restrictions related to the COVID-19 pandemic led to a reduced and irregular school meal offer. Consequently, pupils in both schools often forgot which days food would be served. 


*It may be that they [the classmates] do not remember [the breakfast serving] like me. I often forget it.*
(Boy, 5th grade, GN11)


*Sometimes I have brought a packed lunch with me, that I have forgotten that [food is served in] the canteen and so on, but I usually eat when there is food.*
(Girl, 9th grade, GN1)


*Eh, the challenges are then that [the lunch] is not every day. As I believe (…) that is the predictability. It was discussed that they do not always know what day [lunch is served]. If it had been every day, there would have been more continuity. (…). And then it’s kind of like that it’s safer to bring a packed lunch. Ok, then I know what to eat today. I think he [the son] would have eaten more of the hot food if it had been provided every day. Because there is something about the planning.*
(Mother, n9, GN7)

Predictability also involved wanting to know in advance what kind of food would be served. Pupils could then plan whether they would bring a packed lunch. Many pupils brought a backup packed lunch regardless to ensure that they had food if they did not like what was being served:


*No, the days when there is food at school, prior to it, there is a lot of talk about what it will be tomorrow. Should I make a backup packed lunch, or should I not?*
(Mother, n1, GN5)

Parents at both schools called for more information to better influence their children to attend the school meals:


*If I had known—it is possible I haven’t followed well enough, and I’ve probably not done that. But in terms of pushing my son to attend those meals, I would probably have done it if I had insight into the menu plan and maybe how the food is prepared.*
(Father, n4, GN6)

### 3.5. Conceptual Framework

Based on our data, a conceptual framework was elaborated to conceptualize participation in the school meals (Figure 1). The framework illustrates how participation is not only due to one or more factors but to the interaction among them. Popular food may foster commensality [52]. Moreover, knowing that other classmates gather around the breakfast table or are to be found in the students’ hall can promote the habit of participating in common meals. Regularity of the provision of meals may make pupils and parents being accustomed to substituting the packed lunch with the free meal. These factors may counterbalance the attraction of the shopping center as a ‘competitor’ in participating in the school meals. This study adopted a grounded theory approach, meaning that we started our study without choosing among existing theories, but letting the data promote the development of new conceptualization and choice of suitable theories [47]. This conceptual framework incorporates aspects of already existing theories as to the importance of the ‘perceived behavior of others’—a key element in Bandura’s social cognitive theory or the influence of ‘early adopters’—from the diffusion of innovation theory [53]. Routines as a determinant for behavior have been empathized in theories of practices and previously used for understanding food consumption [54]. By adopting a grounded theory approach, we could shed light on the interconnections between existing theories, new concepts, and the material and cultural context in which participation in school meals is situated. 

## 4. Discussion

This study aimed to understand what influences pupils to participate in free school meal schemes. The findings indicated that simply serving a ‘free meal’ may not be enough to make the children participate. Several concurring factors influence the participation in school meals. Among these, the most relevant ones are the popularity of the food, competition with the nearby shopping center, social aspects, and predictability. The need to look at the interaction of several factors has emerged also in other studies, underling the importance of combining theoretical perspectives looking at individual, social, cultural, and structural factors [55,56,57]. 

The finding that food preferences is one of the main factors influencing pupils’ food choices is in line with several previous studies [39,40,42,58]. A US study on free school breakfast showed that unfulfilled food preferences was a barrier to participation, and the children accepted being hungry rather than eating the school breakfast if they did not like it [39]. In our study, even when pupils participated in the free lunch, it was not given that everything on the plate was eaten. That food waste increases when pupils’ food preferences are not matched is also confirmed by others [59,60]. Our study supports evidence from Finland, where although almost all pupils ate the free meals offered [38], a main challenge was that components such as milk/buttermilk, salad, fresh vegetables, and bread were commonly left uneaten. Furthermore, the most common reason for not participating or eating the meal was that the pupils did not like the food served [38]. 

What was defined as popular food varied. However, as other studies have shown, popular food was seldom associated with healthy food [61,62]. This generated a challenge as a premise of the whole project promoted by Oslo Municipality—and of school meal policies in general—is that the food provided should be healthy. Therefore, finding a balance between providing healthy alternatives without jeopardizing children’s participation in the food schemes emerged as a main challenge of our study. The need to find a balance between different sets of values has also been reported in other studies [63,64]. What was particularly interesting in our study was that not only the adults but also the pupils in charge of preparing the meals were often faced with balancing between priorities and values. Previous articles show that involving children in food preparation, as in our study, helps increase their awareness about healthy and sustainable food and can increase healthy food choices [65,66]. 

The accessibility to a nearby shopping center led to a competition with the free school lunch. These results seem consistent with a study from Finland which found that the availability of grocery stores or fast-food restaurants near schools was associated with higher fast-food purchases [43]. The short distance to these establishments was reversely associated with skipping breakfast and the free school lunch, especially for low-SES adolescents [43]. A rapid evidence review about school food provision in high-income countries identified that many barriers to participation were explicitly related to the school environment and neighborhoods rather than the school meal programs, such as access to competitive foods [58]. As other studies have shown [42,67], going to the mall or fast-food restaurants is part of adolescents’ increasing autonomy. In addition, independently seeking out foods and controlling their own food choices were important issues [42] found related to adolescents’ eating autonomy. 

Eating together and talking with others was the most stated reason for participating in the school breakfast. For some, it meant not having to eat alone at home. Additionally, the free lunch scheme seemed to gather pupils who would otherwise have been alone. That school meals could contribute to preventing loneliness was an essential factor emerging from our study. This finding is particularly important since the Youth in Oslo surveys from 2018 and 2021 have shown an increase in adolescents reporting loneliness [9,68]. Several studies emphasize the importance of eating school meals as a social happening [40,69]. Other studies confirm that free school meals could help develop friendships, improve children’s relationships with other pupils and reduce victimization over time [58,66,69]. A qualitative assessment of free school meals in Norway found several perceived benefits, including social equality, social interaction, and social learning [69]. In general, there are assumptions and evidence that free school meals have the potential to equal out social differences [3,28,70]. However, Andersen et al. note that other social differences may arise when everyone must eat the same food and cultural differences in food habits, ethical and religious aspect, together with other dietary needs are not taken into account [41,71]. One important but not surprising finding in our study was that participation and acceptance of the free meals seemed to depend on what behavior was expected among friends and classmates, as the pupils were influenced by their peers regarding partaking in the free meals. These findings are similar to what was found in a Danish study where there were significant differences in participation between classes, indicating that peers were influenced by their classmates [72]. 

Predictability was another critical aspect influencing participation in school meals. The packed school lunch, traditionally brought from home, has strong roots in Norwegian food culture and identity [73]. Therefore, free school meals represent something the current generation of parents and children are not accustomed to. As indicated in the literature, food consumption is largely a routinized activity, and changes in food consumption practices may need time [64,74]. This is confirmed by the example of the national fruit and vegetable subscription program in Norway, which showed that the introduction of fruit and vegetables in schools took time [75]. Children in an intervention study in Denmark, which assessed pupils’ acceptance of an implemented school lunch, preferred their packed lunches to the provided free hot meals [72]. In our study, free school meals were not only new for pupils; the parents themselves were not accustomed to this new system and called for more regularity and information about the planned menus. Several studies have shown that parents’ perception of a school meal scheme affects their children’s participation in these meals, especially breakfast [76,77]. Therefore, parents’ involvement in free school meals might be beneficial for pupils’ participation [77]. The pilot studies indicate the feasibility and the potential of introducing school meals in schools. However, as suggested by other studies, implementing changes is a complex process influenced by social relationships, routines, and contextual aspects [78].

### Strengths and Limitations 

As provision of meals at school has become a political priority in Norway, studies on the introduction and evaluation of school meals in Norway are increasing [32,34,35,36,37,69]. However, there is a lack of studies including a wide sample of pupils, teachers, and parents. Thus, this study contributes to a better understanding of school meal participation by bringing together different angles.

Elements of the Consolidated Criteria for Reporting Qualitative Research (COREQ) checklist were used to provide detailed information and reflections about the research process [79]. A limitation of the study is that theoretical saturation was not fully achieved. As specified by Charmaz, theoretical saturation in GT occurs when gathering more data sheds no further light on the properties of a theoretical category [80]. The data collected provided a good basis for interpreting and explaining the phenomenon. However, more data could have been gathered to achieve a better understanding of factors related to non-participation in school meals. This limitation may be due to difficulties in recruiting some groups of informants, also because of restrictions related to the COVID-19 pandemic. In line with GT and a theoretical sampling strategy [47], it was desirable to conduct more interviews with those who did not use the scheme, especially with 10th-graders, and parents who were not so interested and engaged in school meals. In addition, despite several attempts, the participants in our study did not reflect the multicultural composition of the pupils in the two schools. Knowing if there are specific barriers that can limit participation in school meals among specific population groups can be of high relevance and, in the perspective of scaling up school meals and make it an opportunity for promoting social equalities in diets [62] attention needs to be given to the groups not reached by this intervention. Another limitation of the study is that part of the focus groups was conducted digitally because of the COVID-19 pandemic. Challenges with web-based interviews as mentioned by other studies were that the researcher could not see the whole body language and the bodily interaction among participants. Furthermore, participants without access to the required technology or feeling uncomfortable using it may have been underrepresented [81]. Problems with the internet connection occurred sometimes during a few of the interviews, which may have distracted participants and led to partial unclear voices on the audio recording.

## 5. Conclusions

To assure that school meals contribute to healthier food habits and wellbeing, high attendance is needed. In Norway, the introduction of school meals is still relatively new. Our study provides findings that can be relevant for the further development and implementation in the Norwegian context and elsewhere. Simply offering free food is not enough. The barriers and motivational aspects presented in this paper should be considered when municipalities and schools plan the implementation of free school meals. To ensure that pupils participate and eat the provided food, schools need to overcome the challenge of meeting children’s food preferences without sacrificing healthiness. Making school meals the preferred option is not however only a matter of taste. Predictability in the provision, better information about the food served, and having a social and friendly eating environment are also important. Regulating access to shopping malls during school time should also be considered, as leaving the school area to purchase snacks and drinks is an option for Norwegian pupils in lower-secondary school. Finally, further research is needed to investigate barriers to participation. 

## Figures and Tables

**Figure 1 nutrients-14-01282-f001:**
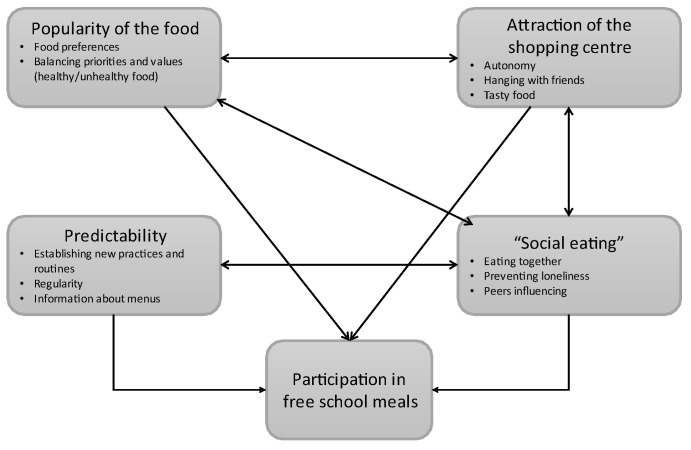
A conceptual framework for participation in school meals.

**Table 1 nutrients-14-01282-t001:** Characteristics of the participants (pupils, parents and school staff) from primary and the lower-secondary schools and number of interviews conducted.

	Number of Participants Total	Number of Interviews Total	Primary School 5th–7th Grade		Lower-Secondary School 8th–10th Grade	
			Participants	Interviews	Participants	Interviews
**Pupils *n***	**39**	**8**	**12**	**5**	**27**	**3**
Boys	12		5		7	
Girls	27		7		20	
Norwegian	24		6		18	
Immigrant	15		6		9	
background						
**Parents *n***	**15**	**7**	**7**	**4**	**8**	**3**
Men	2		1		1	
Women	13		6		7	
Norwegian	13		6		7	
Immigrant	2		1		1	
background						
**School staff *n***	**12**	**5**	**7**	**3**	**5**	**2**
Men	4		3		1	
Women	8		4		4	
**Total**	**66**	**20**	**26**	**12**	**40**	**8**

Bold means total number.

## Data Availability

The transcribed interviews, anonymized, are available from the corresponding author.

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
