# Peer review of "Children’s Participation in Free School Meals: A Qualitative Study among Pupils, Parents, and Teachers"

_nutrients, 2022, doi:10.3390/nu14061282_

Round 1

Reviewer 1 Report

 my suggestions for revision are a little more than minor, but less than major.

Nutrients:  Article Review, February 2022

This type of research is important because it examines practices within schools and the views of relevant groups.  This particular research is based on interviews with a total of 66 people, pupils, parents, and school staff, a significant number of participants, especially during the time of the pandemic.  The article included useful quotes from a variety of participants, which helped the reader gain further insight into this new program.

My comments centre on the research sample, the school observations, data analysis, and the use of grounded theory.

Sample:  while the authors provide us with information about the nature of the sample, there is information that would provide additional context.  First, could the authors clarify how these 66 people became participants relative to those who did not participate (e.g., were all the parents at the school asked to participate and 15 volunteered or . . . )?  That information would provide additional context (although I appreciate that at the end of the article, the authors identified that the participants were not representative of the schools’ population).  Second, do the authors have any comments about data saturation and the variability/similarities among the responses?  Note that in Table 1, something is off with the numbers of Norwegian parents and the immigrant background parents:  12 (6 + 7) and 3 (1 +1)

School observations:  it is very commendable that the researchers visited the school and used these visits to inform the development of the interview guide.  It would be helpful for the authors to provide more information about what they observed and the results of those observations.  For example, did they observe student participation and if so, what was it like; and what about the quality of the food, ambience, food waste, etc.?  Similarly, an example of menus from each program would also provide context for the reader.

Data analysis:  It would be helpful to clarify the roles, if any, of the other researchers in the analysis of the data, aside from discussing the preliminary codebook.  Was there a final codebook?  If so, how was it reached and how was it used?

Grounded theory:  My knowledge of grounded theory is relatively limited but my expectation was that when grounded theory is used as a method, that it is carried throughout the article.  There seemed to be little attention to theoretical aspects of the research in the discussion, such as discussions of other theories or models that could be relevant to this type of research.  When it comes to students’ food preferences, these might include, the Theory of Planned Behaviour, Social Cognitive Theory or the Socio-Ecological Model.  A google scholar search using the phrase – theory of food preferences of school students healthy eating – will identify some articles that may be of interest, including this one:  [HTML] A conceptual framework for healthy eating behavior in Ecuadorian adolescents: a qualitative study

R Verstraeten, K Van Royen, A Ochoa-Avilés… - PloS one, 2014 - journals.plos.org

… To better understand these factors in Ecuadorian adolescents, we used a theoretical …
environment, schools and their children's food preferences as key influences on food choice. This …

Save Cite Cited by 109 Related articles All 24 versions 

]

Likewise, there may be potential merit in considering the introduction of school meal programs as an innovation and/or implementation and thus exploring related conceptual/theoretical models.  In that regard, this review article may be of interest:

Context, complexity and process in the implementation of evidence-based innovation: a realist informed review

KD Dryden-Palmer… - BMC Health …, 2020 - bmchealthservres.biomedcentral …

This review of scholarly work in health care knowledge translation advances understanding
of implementation components that support the complete and timely integration of new
knowledge. We adopt a realist approach to investigate what is known from the current
literature about the impact of, and the potential relationships between, context, complexity
and implementation process. Informed by two distinct pathways, knowledge utilization and
knowledge translation, we utilize Rogers' Diffusion of Innovations theory (DOI) and Harvey …

Save Cite Cited by 24 Related articles All 14 versions 

I provide these examples not because I expect the authors to use them, but only as examples – the authors can decide how to integrate theoretical aspects into the article most appropriately.

Author Response

Thank you for your useful comments. Please find our reply in attachment

Reviewer 2 Report

Overall comments:

Nice study. Paper needs to be checked for grammar and sentence construction.

Specific comments

Lines 14-16: Abstract needs to be checked for sentence construction. The sentence ‘One primary school and one lower-14 secondary school in different districts in Oslo that were implementing two different free school meal 15 models’ is incomplete.

Lines 20-22: ‘Children and parents 20 involvement, regularity of the meals provision, a good flow of information and the creation of a 21 friendly eating environment are recommended’ – this too

Lines 66-68: ‘This was to follow up a…’ Sentence construction is incorrect.

Line 82: clarify who participants are – pupils, staff, and/or parents? Add a note referring to Table 1 here.

Line 117: What was the reason for conducting just one in-depth interview?

Line 381: Could virtual group discussions be limiting as compared to in person focus group discussions where participants and the research team can read body language, expressions, etc?

Line 389: Change ‘Covid restriction.’ To restrictions related to the COVID-19 pandemic.

Lines 407-08: Phrases ‘Regularity in the provision’ and ‘a good flow of information about the food served’ are unclear.

Line 413: ‘meet 412 the aims of among specific groups’ is unclear.

Author Response

Than you for your useful comments. Please find our reply in attachment
